# TARGET TRAINING:
# TRICKING ADVERSARIAL ATTACKS TO FAIL

## ABSTRACT

Recent adversarial defense approaches have failed. Untargeted gradient-based attacks cause classifiers to choose any wrong class. Our novel white-box defense tricks untargeted attacks into becoming attacks targeted at designated target classes. From these target classes, we derive the real classes. The Target Training defense tricks the minimization at the core of untargeted, gradient-based adversarial attacks: minimize the sum of (1) perturbation and (2) classifier adversarial loss. Target Training changes the classifier minimally, and trains it with additional duplicated points (at 0 distance) labeled with designated classes. These differently-labeled duplicated samples minimize both terms (1) and (2) of the minimization, steering attack convergence to samples of designated classes, from which correct classification is derived. Importantly, Target Training eliminates the need to know the attack and the overhead of generating adversarial samples of attacks that minimize perturbations. Without using adversarial samples and against an adaptive attack aware of our defense, Target Training exceeds even default, unsecured classifier accuracy of 84.3% for CIFAR10 with 86.6% against DeepFool attack; and achieves 83.2% against CW-$L_2(\kappa=0)$ attack. Using adversarial samples, we achieve 75.6% against CW-$L_2(\kappa=40)$. Due to our deliberate choice of low-capacity classifiers, Target Training does not withstand $L_\infty$ adaptive attacks in CIFAR10 but withstands CW-$L_\infty(\kappa=0)$ in MNIST. Target Training presents a fundamental change in adversarial defense strategy.

## 1 INTRODUCTION

Neural network classifiers are vulnerable to malicious adversarial samples that appear indistinguishable from original samples (Szegedy et al., 2013), for example, an adversarial attack can make a traffic stop sign appear like a speed limit sign (Eykholt et al., 2018) to a classifier. An adversarial sample created using one classifier can also fool other classifiers (Szegedy et al., 2013; Biggio et al., 2013), even ones with different structure and parameters (Szegedy et al., 2013; Goodfellow et al., 2014; Papernot et al., 2016b; Tramèr et al., 2017b). This transferability of adversarial attacks (Papernot et al., 2016b) matters because it means that classifier access is not necessary for attacks. The increasing deployment of neural network classifiers in security and safety-critical domains such as traffic (Eykholt et al., 2018), autonomous driving (Amodei et al., 2016), healthcare (Faust et al., 2018), and malware detection (Cui et al., 2018) makes countering adversarial attacks important.

Gradient-based attacks use the classifier gradient to generate adversarial samples from non-adversarial samples. Gradient-based attacks minimize at the same time classifier adversarial loss and perturbation (Szegedy et al., 2013), though attacks can relax this minimization to allow for bigger perturbations, for example in the Carlini&Wagner attack (CW) (Carlini & Wagner, 2017c) for $\kappa > 0$, in the Projected Gradient Descent attack (PGD) (Kurakin et al., 2016; Madry et al., 2017), in FastGradientMethod (FGSM) (Goodfellow et al., 2014). Other gradient-based adversarial attacks include DeepFool (Moosavi-Dezfooli et al., 2016), Zeroth order optimization (ZOO) (Chen et al., 2017), Universal Adversarial Perturbation (UAP) (Moosavi-Dezfooli et al., 2017).

Many recent proposed defenses have been broken (Athalye et al., 2018; Carlini & Wagner, 2016; 2017a;b; Tramer et al., 2020). They fall largely into these categories: (1) adversarial sample detection, (2) gradient masking and obfuscation, (3) ensemble, (4) customized loss. Detection defenses (Meng & Chen, 2017; Ma et al., 2018; Li et al., 2019; Hu et al., 2019) aim to detect, cor-

rect or reject adversarial samples. Many detection defenses have been broken (Carlini & Wagner, 2017b;a; Tramer et al., 2020). Gradient obfuscation is aimed at preventing gradient-based attacks from access to the gradient and can be achieved by shattering gradients (Guo et al., 2018; Verma & Swami, 2019; Sen et al., 2020), randomness (Dhillon et al., 2018; Li et al., 2019) or vanishing or exploding gradients (Papernot et al., 2016a; Song et al., 2018; Samangouei et al., 2018). Many gradient obfuscation methods have also been successfully defeated (Carlini & Wagner, 2016; Athalye et al., 2018; Tramer et al., 2020). Ensemble defenses (Tramèr et al., 2017a; Verma & Swami, 2019; Pang et al., 2019; Sen et al., 2020) have also been broken (Carlini & Wagner, 2016; Tramer et al., 2020), unable to even outperform their best performing component. Customized attack losses defeat defenses (Tramer et al., 2020) with customized losses (Pang et al., 2020; Verma & Swami, 2019) but also, for example ensembles (Sen et al., 2020). Even though it has not been defeated, Adversarial Training (Kurakin et al., 2016; Szegedy et al., 2013; Madry et al., 2017) assumes that the attack is known in advance and takes time to generate adversarial samples at every iteration. The inability of recent defenses to counter adversarial attacks calls for new kinds of defensive approaches.

In this paper, we make the following major contributions:

- We develop Target Training - a novel, white-box adversarial defense that converts untargeted gradient-based attacks into attacks targeted at designated target classes, from which correct classes are derived. Target Training is based on the minimization at the core of untargeted gradient-based adversarial attacks.

- For all attacks that minimize perturbation, we eliminate the need to know the attack or to generate adversarial samples during training.

- We show that Target Training withstands non-$L_\infty$ adversarial attacks without resorting to increased network capacity. With default accuracy of 84.3% in CIFAR10, Target Training achieves 86.6% against the DeepFool attack, and 83.2% against the CW-$L_2$($\kappa$=0) attack without using adversarial samples and against an adaptive attack aware of our defense. Against an adaptive CW-$L_2$($\kappa$=40) attack, we achieve 75.6% while using adversarial samples. Our choice of low-capacity classifiers makes Target Training not withstand $L_\infty$ adaptive attacks, except for CW-$L_\infty$($\kappa$=0) in MNIST.

- We conclude that Adversarial Training might not be defending by populating sparse areas with samples, but by minimizing the same minimization that Target Training minimizes.

## 2 BACKGROUND AND RELATED WORK

Here, we present the state-of-the-art in adversarial attacks and defenses, as well as a summary.

**Notation** A $k$-class neural network classifier that has $\theta$ parameters is denoted by a function $f(x)$ that takes input $x \in \mathbb{R}^d$ and outputs $y \in \mathbb{R}^k$, where $d$ is the dimensionality and $k$ is the number of classes. An adversarial sample is denoted by $x_{adv}$. Classifier output is $y$, $y_i$ is the probability that the input belongs to class $i$. Norms are denoted as $L_0$, $L_2$ and $L_\infty$.

### 2.1 ADVERSARIAL ATTACKS

Szegedy et al. (2013) were first to formulate the generation of adversarial samples as a constrained minimization of the perturbation under an $L_p$ norm. Because this formulation can be hard to solve, Szegedy et al. (2013) reformulated the problem as a gradient-based, two-term minimization of the weighted sum of perturbation, and classifier loss. For untargeted attacks, this minimization is:

$$\begin{aligned} \text{minimize} \quad & c \cdot \|x_{adv} - x\|_2^2 + loss_f(x_{adv}) \quad & \text{(Minimization 1)} \\ \text{subject to} \quad & x_{adv} \in [0,1]^n \end{aligned}$$

where $f$ is the classifier, $loss_f$ is classifier loss on adversarial input, and $c$ a constant value evaluated in the optimization. Term (1) is a norm to ensure a small adversarial perturbation. Term (2) utilizes the classifier gradient to find adversarial samples that minimize classifier adversarial loss.

Minimization 1 is the foundation for many gradient-based attacks, though many tweaks can and have been applied. Some attacks follow Minimization 1 implicitly (Moosavi-Dezfooli et al., 2016),

and others explicitly (Carlini & Wagner, 2017c). The type of $L_p$ norm in term (1) of the minimization also varies. For example the CW attack (Carlini & Wagner, 2017c) uses $L_0$, $L_2$ and $L_\infty$, whereas DeepFool (Moosavi-Dezfooli et al., 2016) uses the $L_2$ norm. A special perturbation case is the Pixel attack by Su et al. (2019) which changes exactly one pixel. Some attacks even exclude term (1) from the Minimization 1 and introduce an external parameter to control perturbation. The FGSM attack by Goodfellow et al. (2014), for example, uses an $\epsilon$ parameter, while the CW attack (Carlini & Wagner, 2017c) uses a $\kappa$ confidence parameter.

The Fast Gradient Sign Method by Goodfellow et al. (2014) is a simple, $L_\infty$-bounded attack that constructs adversarial samples by perturbing each input dimension in the direction of the gradient by a magnitude of $\epsilon$: $x_{adv} = x + \epsilon \cdot sign(\nabla_x loss(\theta, x, y))$.

The current strongest attack is CW (Carlini & Wagner, 2017c). CW customizes Minimization 1 by passing $c$ to the second term, and using it to tune the relative importance of the terms. With a further change of variable, CW obtains an unconstrained minimization problem that allows it to optimize directly through back-propagation. In addition, CW has a $\kappa$ parameter for controlling the confidence of the adversarial samples. For $\kappa > 0$ and up to 100, the CW attack allows for more perturbation in the adversarial samples it generates.

The DeepFool attack by Moosavi-Dezfooli et al. (2016) follows the Minimization 1 implicitly. DeepFool (Moosavi-Dezfooli et al., 2016) looks at the smallest distance of a point from the classifier decision boundary as the minimum amount of perturbation needed to change its classification. DeepFool approximates the classifier with a linear one, estimates the distance from the linear boundary, and then takes steps in the direction of the closest boundary until an adversarial sample is found.

**Black-box attacks** Black-box attacks assume no access to classifier gradients. Such attacks with access to output class probabilities are called score-based attacks, for example the ZOO attack (Chen et al., 2017), a black-box variant of the CW attack (Carlini & Wagner, 2017c). Attacks with access to only the final class label are decision-based attacks, for example the Boundary (Brendel et al., 2017) and the HopSkipJumpAttack (Chen et al., 2019) attacks.

**Multi-step attacks** The PGD attack (Kurakin et al., 2016) is an iterative method with an $\alpha$ parameter that determines a step-size perturbation magnitude. PGD starts at a random point $x_0$, projects the perturbation on an $L_p$-ball $B$ at each iteration: $x(j + 1) = Proj_B(x(j) + \alpha \cdot sign(\nabla_x loss(\theta, x(j), y))$. The BIM attack (Kurakin et al., 2016) applies FGSM (Goodfellow et al., 2014) iteratively with an $\alpha$ step. To find a universal perturbation, UAP (Moosavi-Dezfooli et al., 2017) iterates over the images and aggregates perturbations calculated as in DeepFool.

## 2.2 ADVERSARIAL DEFENSES

**Adversarial Training.** Adversarial Training (Szegedy et al., 2013; Kurakin et al., 2016; Madry et al., 2017) is one of the first and few, undefeated defenses. It defends by populating low probability, so-called blind spots (Szegedy et al., 2013; Goodfellow et al., 2014) with adversarial samples labelled correctly, redrawing boundaries. The drawback of Adversarial Training is that it needs to know the attack in advance, and it needs to generate adversarial samples during training. The Adversarial Training algorithm 2 in the Appendix is based on Kurakin et al. (2016). Madry et al. (2017) formulate their defense as a robust optimization problem, and use adversarial samples to augment the training. Their solution however necessitates high-capacity classifiers - bigger models with more parameters.

**Detection defenses** Such defenses detect adversarial samples implicitly or explicitly, then correct or reject them. So far, many detection defenses have been defeated. For example, ten diverse detection methods (other network, PCA, statistical properties) were defeated by attack loss customization (Carlini & Wagner, 2017a); Tramer et al. (2020) used attack customization against (Hu et al., 2019); attack transferability (Carlini & Wagner, 2017b) was used against MagNet by Meng & Chen (2017); deep feature adversaries (Sabour et al., 2016) against (Roth et al., 2019).

**Gradient masking and obfuscation** Many defenses that mask or obfuscate the classifier gradient have been defeated (Carlini & Wagner, 2016; Athalye et al., 2018). Athalye et al. (2018) identify three types of gradient obfuscation: (1) Shattered gradients - incorrect gradients caused by non-differentiable components or numerical instability, for example with multiple input transformations by Guo et al. (2018). Athalye et al. (2018) counter such defenses with Backward Pass Differentiable

Approximation. (2) Stochastic gradients in randomized defenses are overcome with Expectation Over Transformation (Athalye et al., 2017) by Athalye *et al*. Examples are Stochastic Activation Pruning (Dhillon et al., 2018), which drops layer neurons based on a weighted distribution, and (Xie et al., 2018) which adds a randomized layer to the classifier input. (3) Vanishing or exploding gradients are used, for example, in Defensive Distillation (DD) (Papernot et al., 2016a) which reduces the amplitude of gradients of the loss function. Other examples are PixelDefend (Song et al., 2018) and Defense-GAN (Samangouei et al., 2018). Vanishing or exploding gradients are broken with parameters that avoid vanishing or exploding gradients (Carlini & Wagner, 2016).

**Complex defenses** Defenses combining several approaches, for example (Li et al., 2019) which uses detection, randomization, multiple models and losses, can be defeated by focusing on the main defense components (Tramer et al., 2020).(Verma & Swami, 2019; Pang et al., 2019; Sen et al., 2020) are defeated ensemble defenses combined with numerical instability (Verma & Swami, 2019), regularization (Pang et al., 2019), or mixed precision on weights and activations (Sen et al., 2020).

## 2.3 SUMMARY

Many defenses have been broken. They focus on changing the classifier. Instead, our defense changes the classifier minimally, but forces attacks to change convergence. Target Training is the first defense based on Minimization 1 at the core of untargeted gradient-based adversarial attacks.

## 3 TARGET TRAINING

Target Training converts untargeted attacks to attacks targeted at designated classes, from which correct classification is derived. Untargeted gradient-based attacks are based on Minimization 1 (on page 2) of the sum of (1) perturbation and (2) classifier adversarial loss. Target Training trains the classifier with samples of designated classes that minimize both terms of the minimization at the same time. These samples are exactly what adversarial attacks look for, based on Minimization 1: nearby points (at 0 distance) that minimize adversarial loss. For attacks that relax the minimization by removing the perturbation term, we adjust Target Training to use adversarial samples against attacks that do not minimize perturbation.

Target Training eliminates the need to know the attack or to generate adversarial samples against attacks that minimize perturbation. Instead of adversarial samples, we use original samples labelled as designated classes because they have minimum 0-distance from original samples. Following, we outline how Target Training minimizes both terms of Minimization 1 simultaneously.

**Term (1) of the minimization - perturbation.** Against attacks that *do minimize* perturbation, such as CW ($\kappa = 0$) and DeepFool, Target Training uses duplicates of original samples in each batch instead of adversarial samples because these samples minimize the perturbation to 0 - no other points can have smaller distance. This eliminates the overhead of calculating adversarial samples against all attacks that minimize perturbation. Algorithm 1 shows Target Training against attacks that minimize perturbations. Against attacks that do *not* minimize perturbation, such as CW ($\kappa > 0$), PGD and FGSM, Target Training replaces duplicated samples with adversarial samples from the attack. The adjusted algorithm is shown in Algorithm 3 in the Appendix.

---

**Algorithm 1** Target Training of classifier $N$ against attacks that minimize perturbation

---

**Require:** Batch size is $m$, number of dataset classes is $k$, untrained classifier $N$ with $2k$ output classes, TRAIN trains a classifier on a batch and its ground truth
**Ensure:** Classifier $N$ is Target-Trained against all attacks that minimize perturbation
    **while** training not converged **do**
        $B = \{x^1, ..., x^m\}$                                                ▷ Get random batch
        $G = \{y^1, ..., y^m\}$                                      ▷ Get batch ground truth
        $B' = \{x^1, ..., x^m, x^1, ..., x^m\}$                        ▷ Duplicate batch
        $G' = \{y^1, ..., y^m, y^1 + k, ..., y^m + k\}$ ▷ Duplicate ground truth and increase duplicates by $k$
        TRAIN($N, B', G'$)           ▷ Train classifier on duplicated batch and new ground truth
    **end while**

---

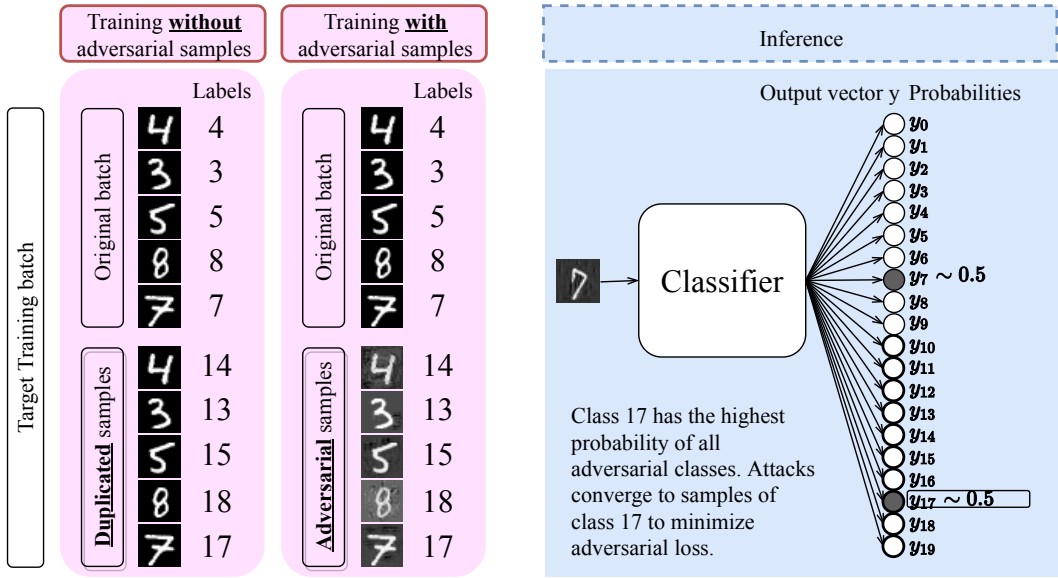

Figure 1: Target Training with and without adversarial samples, and output probabilities at inference. Example images are from the MNIST dataset, smaller batch size shown for brevity. Actual inference output probability values are shown in the Appendix in Table 8 for MNIST, in Table 9 for CIFAR10.

**Term (2) of the minimization - classifier adversarial loss.** In a converged, multi-class classifier, the probabilities of adversarial classes are $\sim 0$ with no distinction among them (the real class has high $\sim 1$ probability). This causes untargeted attacks to converge to samples of whichever highest probability adversarial class, due to the minimization of adversarial loss in term (2). In order to force attack convergence to designated classes, a classifier would need to train so that one of the adversarial classes has higher probability than the other adversarial classes. Which means that the classifier would need to have two high probability outputs: the real class, and the designated adversarial class.

**Target Training.** Against attacks that minimize perturbation, Target Training duplicates training samples in each batch and labels duplicates with designated classes. Against attacks that do not minimize perturbation, adversarial samples are used instead of duplicates. As a result, the real class and the designated class have equal inference probabilities of $\sim 0.5$, as shown in Figure 1. Since attacks minimize adversarial loss in term (2) of Minimization 1, attacks converge to adversarial samples from the designated class. The same samples also minimize term (1) because the classifier was trained with: samples with 0 distance, or adversarial samples close to original samples. It does not affect Target Training whether the $c$ constant of the Minimization 1 is at term (1) or term (2) in an attack because both terms are minimized. Target Training could be extended to defend simultaneously against many attacks using a designated class for each type of attack. During training, adversarial samples would be labeled with the corresponding designated class. In this paper, we do not conduct experiments on simultaneous multi-attack defense.

**Model structure and inference.** The only change to classifier structure is doubling the number of output classes from $k$ to $2k$, loss function remains standard softmax cross entropy. Inference calculation is: $C(x) = \arg\max_i (y_i + y_{i+k}), i \in [0 \ldots (k-1)]$.

## 4 EXPERIMENTS AND RESULTS

**Threat model** We assume that the *adversary goal* is to generate adversarial samples that cause untargeted misclassification. We perform white-box evaluations, assuming the adversary has complete *knowledge* of the classifier and how the defense works. In terms of *capabilities*, we assume

that the adversary is gradient-based, has access to the CIFAR10 and MNIST image domains and is able to manipulate pixels. Against attacks that minimize perturbations, no adversarial samples are used in training and no further assumptions are made about attacks. For attacks that do not minimize perturbations, we assume that the attack is of the same kind as the attack used to generate the adversarial samples used during training. Perturbations are assumed to be $L_p$-constrained.

**Attack parameters** For *PGD*, we use parameters based on the PGD paper Madry et al. (2017): 7 steps of size 2 with a total $\epsilon = 8$ for CIFAR10, and 40 steps of size 0.01 with a total $\epsilon = 0.3$ for MNIST. For all PGD attacks, we use 0 random initialisations within the $\epsilon$ ball, effectively starting PGD attacks from the original images. For *CW*, we use 1000 iterations by default but we also run experiments with up to 100000 iterations. Confidence values in CW range from 0 to 40. For the *ZOO* attack, we use parameters based on the ZOO attack paper (Chen et al., 2017): 9 binary search steps, 1000 iterations, initial constant $c = 0.01$. Additionally, for the ZOO attack, we use 200 randomly selected images from MNIST and CIFAR10 test sets, same as is done in ZOO (Chen et al., 2017) for untargeted attacks. For *FGSM*, $\epsilon = 0.3$, as in (Madry et al., 2017).

**Classifier models** We purposefully do not use high capacity models, such as ResNet (He et al., 2016), used for example by Madry et al. (2017). The reason is to show that Target Training does not necessitate high model capacity to defend against adversarial attacks. The architectures of MNIST and CIFAR datasets are shown in Table 5 in the Appendix, and no data augmentation was used. Default accuracies without attack are 99.1% for MNIST and 84.3% for CIFAR10.

**Datasets** The MNIST (LeCun et al., 1998) and the CIFAR10 (Krizhevsky et al., 2009) datasets are 10-class datasets that have been used throughout previous work. The MNIST (LeCun et al., 1998) dataset has $60K$, $28 \times 28 \times 1$ hand-written, digit images. The CIFAR10 (Krizhevsky et al., 2009) dataset has $70K$, $32 \times 32 \times 3$ images. All experimental evaluations are with testing samples.

**Tools** Adversarial samples generated with CleverHans 3.0.1 (Papernot et al., 2018) for CW (Carlini & Wagner, 2017c), DeepFool (Moosavi-Dezfooli et al., 2016), FGSM (Goodfellow et al., 2014) attacks and IBM Adversarial Robustness 360 Toolbox (ART) toolbox 1.2 (Nicolae et al., 2018) for other attacks. Target Training is written in Python 3.7.3, using Keras 2.2.4 (Chollet et al., 2015).

### 4.1 TARGET TRAINING AGAINST ATTACKS THAT MINIMIZE PERTURBATION

We compare Target Training with unsecured classifier as well as Adversarial Training, since other defenses have been defeated (Carlini & Wagner, 2017b;a; 2016; Athalye et al., 2018; Tramer et al., 2020) successfully. Table 1 shows that, while not using adversarial samples against adversarial attacks that minimize perturbation, Target Training exceeds by far accuracies by unsecured classifier and Adversarial Training in both CIFAR10 and MNIST. In CIFAR10, Target Training even exceeds the accuracy of the unsecured classifier on non-adversarial samples (84.3%) for most attacks.

### 4.2 TARGET TRAINING AGAINST ATTACKS THAT DO NOT MINIMIZE PERTURBATION

Against adversarial attacks that do not minimize perturbation, Target Training uses adversarial samples and performs slightly better than Adversarial Training. We choose Adversarial Training as a baseline as an undefeated adversarial defense, more details in Section 2. Our implementation of Adversarial Training is based on (Kurakin et al., 2016), shown in Algorithm 2. In Table 2, we show that Target Training performs slightly better than Adversarial Training against attacks that do not minimize perturbation.

In Table 6 in the Appendix, we show that both Target Training and Adversarial Training are able to defend in some cases against attacks, the adversarial samples of which have not been used during training - Target Training in more cases than Adversarial Training.

### 4.3 TARGET TRAINING PERFORMANCE ON ORIGINAL, NON-ADVERSARIAL SAMPLES

In Table 7 in the Appendix, we show that Target Training exceeds default classifier accuracy on original, non-adversarial images when trained without adversarial samples against attacks that minimize perturbation: 86.7% (up from 84.3%) in CIFAR10. Furthermore, Table 7 shows that when

Table 1: Here, we show Target Training performance against attacks that minimize perturbation, for which Target Training does not use adversarial samples. Target Training exceeds performance of default, unsecured classifier. Target Training also exceeds the performance of classifier trained with Adversarial Training that uses adversarial samples. For several attacks, Target Training even exceeds the accuracy of default classifier on non-adversarial samples in CIFAR10 (84.3%).

| Attack | CIFAR10 (84.3%) | | | MNIST (99.1%) | | |
|---|---|---|---|---|---|---|
| | Target Training | Advers. Training | Default Classifier | Target Training | Advers. Training | Default Classifier |
| CW-$L_2$, $\kappa = 0$, 1K iter. | 85.6% | 22.8% | 8.8% | 96.3% | 5.0% | 0.8% |
| DeepFool | 86.6% | 24.0% | 9.2% | 94.9% | 5.2% | 1.3% |
| ZOO$^\star$ | 89.0% | - | 81.5% | 93.0% | - | 96.0% |
| UAP | 86.8% | 42.5% | 17.2% | 98.6% | 58.4% | 42.1% |
| CW-$L_\infty$, $\kappa = 0$, 1K iter. | 84.2% | 21.4% | 42.0% | 96.3% | 80.1% | 82.1% |

$^\star$ Based on ZOO paper (Chen et al., 2017), 200 randomly selected test images are used for the untargeted ZOO attack. Accuracies for Adversarial Training against ZOO attack are not calculated because the ZOO attack is slow in generating the adversarial samples needed by Adversarial Training.

Table 2: Using adversarial samples in training, Target Training performs slightly better than Adversarial Training against attacks that do not minimize perturbation.

| Attack | CIFAR10 (84.3%) | | | MNIST (99.1%) | | |
|---|---|---|---|---|---|---|
| | Target Training | Adversarial Training | Unsecured Classifier | Target Training | Adversarial Training | Unsecured Classifier |
| CW-$L_2(\kappa = 40)$ | 77.7% | 77.4% | 8.5% | 98.0% | 98.0% | 0.7% |
| PGD | 76.3% | 76.2% | 32.7% | 92.3% | 91.7% | 79.7% |
| FGSM($\epsilon = 0.3$) | 72.1% | 71.8% | 11.8% | 98.0% | 98.4% | 10.0% |

using adversarial samples against attacks that do not minimize perturbation, Target Training equals Adversarial Training performance on original, non-adversarial images. This holds for both MNIST and CIFAR10.

## 4.4 SUMMARY OF RESULTS

Our Section 4.1 experiments show that we substantially improve performance against attacks that minimize perturbation without using adversarial samples, surpassing even default accuracy for CIFAR10. Section 4.2 experiments show that against attacks that do not minimize perturbation, Target Training has slightly better performance than the current non-broken defense, Adversarial Training.

## 4.5 TRANSFERABILITY ANALYSIS

For a defense to be strong, it needs to be shown to break the transferability of attacks. A good source of adversarial samples for transferability is the unsecured classifier (Carlini et al., 2019). We experiment on the transferability of attacks from the unsecured classifier to a classifier secured with Target Training. In Table 3, we show that Target Training breaks the transferability of adversarial samples generated by attacks that minimize perturbation. Against attacks that do not minimize perturbation, the transferability is broken in MNIST, and partially in CIFAR10.

## 5 ADAPTIVE EVALUATION

Many recent defenses have failed to anticipate attacks that have defeated them (Carlini et al., 2019; Carlini & Wagner, 2017a; Athalye et al., 2018). To avoid that, we perform an adaptive evaluation (Carlini et al., 2019; Tramer et al., 2020) of our Target Training defense.

Table 3: Target Training breaks the transferability of attacks generated from unsecured classifier by maintaining high accuracy against attacks generated using the unsecured classifier in attacks that minimize perturbation for CIFAR10 and MNIST. For attacks that do not minimize perturbation, Target Training breaks the transferability in MNIST, and partially in CIFAR10.

| | CIFAR10 (84.3%) | | MNIST (99.1%) | |
|---|---|---|---|---|
| Attack | Target Training | Unsecured Classifier | Target Training | Unsecured Classifier |
| CW-$L_2$($\kappa = 0$), 1K iter. | 69.9% | 8.8% | 78.3% | 0.8% |
| CW-$L_\infty$($\kappa = 0$), 1K iter. | 76.6% | 42.0% | 93.5% | 82.1% |
| DeepFool | 74.8% | 9.2% | 96.5% | 1.3% |
| CW-$L_2$($\kappa = 40$), 1K iter. | 34.7% | 8.5% | 95.1% | 0.7% |
| PGD | 36.8% | 32.7% | 92.2% | 79.7% |

**Whether Target Training could be defeated by methods used to break other defenses.** Adaptive attack approaches (Carlini & Wagner, 2017b;a; 2016; Athalye et al., 2018; Tramer et al., 2020) used to defeat most current defenses cannot break Target Training defense because we use none of the previous defense approaches: adversarial sample detection, preprocessing, obfuscation, ensemble, customized loss, subcomponent, non-differentiable component, or special model layers. We also keep the loss function simple - standard softmax cross-entropy and no additional loss.

**Adaptive attack against Target Training.** We consider an adversarial attack that is aware that a classifier uses Target Training. The attack adds a new layer at the top of the classifier, with 20 inputs and 10 outputs, where output $i$ is the sum of input $i$ and $i + k$. The classifier with the extra layer is used to generate adversarial samples that are tested on the original classifier. We summarize the results of the adaptive attack in Table 4.

Table 4: Target Training performance against adaptive adversarial attack, for CIFAR10 and MNIST. Adversarial images are generated using the extra-layer classifier and tested against the original Target-Trained classifier. Target Training withstands the adaptive attack when the norm is not $L_\infty$. When the norm is $L_\infty$, Target Training withstands the attack only for MNIST CW-$L_\infty$($\kappa = 0$).

| | CIFAR10 (84.3%) | | MNIST (99.1%) | |
|---|---|---|---|---|
| Attack | Target Training | Unsecured Classifier | Target Training | Unsecured Classifier |
| CW-$L_2$($\kappa = 0$), 1K iter. | 83.2% | 8.8% | 95.7% | 0.8% |
| DeepFool | 86.6% | 9.2% | 94.9% | 1.3% |
| CW-$L_\infty$($\kappa = 0$), 1K iter. | 20.8% | 42.0% | 95.7% | 82.1% |
| CW-$L_2$($\kappa = 40$), 1K iter. | 75.6% | 8.5% | 93.9% | 0.7% |
| PGD | 7.1% | 32.7% | 57.9% | 79.7% |

Table 4 shows that Target Training withstands the adaptive attack when the norm of the attack is not $L_\infty$, even surpassing default classifier accuracy on non-adversarial images for DeepFool ($L_2$) in CIFAR10. For $L_\infty$-based attacks, PGD and CW-$L_\infty$($\kappa = 0$), Target Training performace decreases. Withstanding $L_\infty$ attacks has previously been shown by Madry et al. (2017) to require high-capacity architecture in classifiers. We suspect that the low capacity of our classifiers causes Target Training to not defend the classifiers against $L_\infty$ attacks. The capacity of our classifiers was deliberately chosen to be low in order to investigate whether Target Training can defend low-capacity classifiers.

*How does Target Training withstand this adaptive attack when the norm is not $L_\infty$?* The addition of the extra layer to the Target-Trained classifier effectively converts the classifier into an Adversarial-Trained classifier: the classifier was originally trained with additional nearby samples that are now labelled correctly. As a result, the adversarial samples generated from the extra-layer classifier are non-adversarial, as in Adversarial Training, and classify correctly in the original classifier. Therefore, what makes Adversarial Training work, also makes Target Training work in this adaptive attack.

**Other adaptive attack against Target Training.**   Most current untargeted attacks, including the strongest one, CW, are based on Minimization 1. But this minimization is a simplification of the hard problem of generating adversarial attacks, as outlined by Szegedy et al. (2013). In order to defeat Target Training, attacks would need to solve the much harder problem of generating adversarial attacks without using gradients. This remains an open problem.

**Iterative attacks.**   The multi-step PGD (Kurakin et al., 2016) attack decreases Target Training accuracy more than single-step attacks, which suggests that our defense is working correctly, according to Carlini et al. (2019).

**Transferability.**   We conduct transferability attacks to show that Target Training can withstand attacks that use another classifier to generate adversarial samples. Table 3 shows that Target Training breaks the transferability of attacks generated using an unsecured classifier, as defenses should according to Carlini et al. (2019).

**Stronger CW attack leads to better Target Training accuracy.**   Many attacks fail to defend from attacks when attacks become stronger. Increasing iterations for CW-$L_2(\kappa = 0)$ 100-fold from $1K$ to $100K$ increases our defense's accuracy. In CIFAR10, the accuracy increases from $85.6\%$ to $86.2\%$, in MNIST from $96.3\%$ to $96.6\%$. Target Training accuracy increases because higher number of iterations means that the attack converges better to samples of the designated classes, leading to higher accuracy.

**No gradient masking or obfuscation.**   The fact that Target Training defends against black-box ZOO attack is also an indicator that Target Training does not do gradient masking or obfuscation according to Carlini et al. (2019).

## 6   DISCUSSION AND CONCLUSIONS

In conclusion, Target Training enables low-capacity classifiers to defend against non-$L_\infty$ attacks, in the case of attacks that minimize perturbation even without using adversarial samples. Target Training achieves this by replacing adversarial samples with original samples, which minimize the adversarial perturbation better. This eliminates the need to know the attack in advance, and the overhead of adversarial samples, for all attacks that minimize perturbation. In contrast, the previous non-broken Adversarial Training defense needs to know the attack and to generate adversarial samples of the attack during training. In addition, Target Training minimizes adversarial loss using designated classes. Our experiments show that Target Training can even exceed default accuracy on non-adversarial samples in CIFAR10, when against non-$L_\infty$ adaptive attacks that are aware of Target Training defense. We attribute $L_\infty$ adaptive attack performance to low classifier capacity. In CIFAR10, Target Training achieves $86.6\%$ against the adaptive DeepFool attack without using adversarial samples, exceeding default accuracy of $84.3\%$. Against the CW-$L_2(\kappa{=}0)$ adaptive attack and without using adversarial samples, Target Training achieves $83.2\%$. Against adaptive CW-$L_2(\kappa{=}40)$ attack, we achieve $75.6\%$ while using adversarial samples.

Target Training resilience to non-$L_\infty$ adaptive attacks can offer a different explanation for how Adversarial Training defends classifiers. The commonly-accepted explanation is that Adversarial Training defends by populating low-density areas with adversarial samples labeled correctly. However, the adaptive attack in Section 5 fails for CW-$L_2(\kappa = 0)$ and DeepFool attacks when using what has become an Adversarial-Trained classifier trained with only original samples instead of adversarial samples. The failure of the adaptive attack when using this Adversarial-Trained classifier trained without adversarial samples raises a question. Might it be that Adversarial Training defends not by filling out the space with more adversarial samples labeled correctly, but in the same way that Target Training does: by minimizing the terms of Minimization 1? If the answer were yes, it would also explain the similarity of the results of Target Training and Adversarial Training against attacks that do not minimize perturbation.

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

## A    APPENDIX

---

**Algorithm 2** Adversarial Training of classifier $N$

---

**Require:** Batch size is $m$, number of dataset classes is $k$, untrained classifier $N$ with $k$ output classes, ADV_ATTACK is an adversarial attack, TRAIN trains classifier on a batch and its ground truth

**Ensure:** Adversarially-Trained classifier $N$

    **while** training not converged **do**

        $B = \{x^1, ..., x^m\}$                                   ▷ Get random batch

        $G = \{y^1, ..., y^m\}$                               ▷ Get batch ground truth

        $A = ADV\_ATTACK(N, B)$                ▷ Generate adv. samples from batch

        $B' = B \bigcup A = \{x^1, ..., x^m, x^1_{adv}, ..., x^m_{adv}\}$        ▷ New batch

        $G' = \{y^1, ..., y^m, y^1, ..., y^m\}$              ▷ Duplicate ground truth

        TRAIN$(N, B', G')$        ▷ Train classifier on new batch and new ground truth

    **end while**

---

Table 5: Architectures of Target Training classifiers for CIFAR10 and MNIST datasets. For the convolutional layers, we use $L_2$ kernel regularizer. Notice that the final Dense.Softmax layers in both models have 20 output classes, twice the number of dataset classes. The default, unsecured classifiers have the same architectures, except the final layers have 10 output classes: Dense.Softmax 10.

| CIFAR10 | MNIST |
|---|---|
| Conv.ELU 3x3x32 | Conv.ReLU 3x3x32 |
| BatchNorm | BatchNorm |
| Conv.ELU 3x3x32 | Conv.ReLU 3x3x64 |
| BatchNorm | BatchNorm |
| MaxPool 2x2 | MaxPool 2x2 |
| Dropout 0.2 | Dropout 0.25 |
| Conv.ELU 3x3x64 | Dense 128 |
| BatchNorm | Dropout 0.5 |
| Conv.ELU 3x3x64 | Dense.Softmax **20** |
| BatchNorm | |
| MaxPool 2x2 | |
| Dropout 0.3 | |
| Conv.ELU 3x3x128 | |
| BatchNorm | |
| Conv.ELU 3x3x128 | |
| BatchNorm | |
| MaxPool 2x2 | |
| Dropout 0.4 | |
| Dense.Softmax **20** | |

---

**Algorithm 3** Target Training of classifier $N$ using adversarial samples against attacks that do not minimize perturbation

---

**Require:** Batch size is $m$, number of dataset classes is $k$, untrained classifier $N$ with $2k$ output classes, ADV_ATTACK is an adversarial attack, TRAIN trains classifier on a batch and its ground truth

**Ensure:** Classifier $N$ is Target-Trained against ADV_ATTACK

    **while** training not converged **do**

        $B = \{x^1, ..., x^m\}$                                   ▷ Get random batch

        $G = \{y^1, ..., y^m\}$                               ▷ Get batch ground truth

        $A = ADV\_ATTACK(N, B)$                ▷ Generate adv. samples from batch

        $B' = B \bigcup A = \{x^1, ..., x^m, x^1_{adv}, ..., x^m_{adv}\}$    ▷ Assemble new batch from original batch and adversarial samples

        $G' = \{y^1, ..., y^m, y^1 + k, ..., y^m + k\}$ ▷ Duplicate ground truth and increase duplicates by $k$

        TRAIN$(N, B', G')$        ▷ Train classifier on new batch and new ground truth

    **end while**

---

Table 6: Expanded comparison of performance against attacks that do not minimize perturbation shows that both Target Training and Adversarial Training can defend against attacks, the adversarial samples of which have not been used in training. Bold font has been used to highlight where such defense was effective. Both Target Training and Adversarial Training defend against some attacks that they have not been trained for, but not all. Unsecured classifier performance also provided as a performance baseline. Target Training appears to be slightly better at defending against attacks, the samples of which have not been used in training.

| Adv. samples in training | Adv. samples in testing | CIFAR10 (84.3%) | | | MNIST (99.1%) | | |
|---|---|---|---|---|---|---|---|
| | | Target Train. | Advers. Train. | No Def. | Target Train. | Adv. Train. | No Def. |
| *CW-$L_2$* *(conf=40)* | *CW-$L_2$ ($\kappa = 40$)* | **77.7%** | **77.4%** | 8.5% | **98.0%** | **98.0%** | 0.7% |
| | CW-$L_2$ ($\kappa = 0$) | **71.3%** | 12.3% | 8.8% | **97.4%** | 1.5% | 8.8% |
| | DeepFool | **75.8%** | 13.2% | 9.2% | **97.6%** | 1.6% | 1.3% |
| | PGD | 10.0% | 10.0% | 32.7% | 23.3% | 2.9% | 79.7% |
| | FGSM($\epsilon = 0.3$) | 10.6% | 9.9% | 11.8% | **56.6%** | 15.8% | 10.0% |
| | FGSM($\epsilon = 0.01$) | 48.9% | 36.4% | 40.4% | **97.7%** | **97.8%** | 98.6% |
| *PGD* | *PGD* | **76.3%** | **76.2%** | 32.7% | **92.3%** | **91.7%** | 79.7% |
| | CW-$L_2$ ($\kappa = 40$) | 7.3% | **57.3%** | 8.5% | **83.2%** | **98.4%** | 0.7% |
| | CW-$L_2$ ($\kappa = 0$) | 12.8% | 12.7% | 8.8% | **94.3%** | 22.7% | 8.8% |
| | DeepFool | 15.0% | 13.0% | 9.2% | **86.5%** | 4.7% | 1.3% |
| | FGSM($\epsilon = 0.3$) | 10.7% | 10.2% | 11.8% | **79.9%** | **95.4%** | 10.0% |
| | FGSM($\epsilon = 0.01$) | 39.8% | 41.5% | 40.4% | **98.2%** | **98.4%** | 98.6% |
| *FGSM* *($\epsilon = 0.3$)* | *FGSM($\epsilon = 0.3$)* | **72.1%** | **71.8%** | 11.8% | **98.0%** | **98.4%** | 10.0% |
| | FGSM($\epsilon = 0.01$) | 40.8% | 42.1% | 40.4% | **98.5%** | **98.5%** | 98.6% |
| | CW-$L_2$ ($\kappa = 40$) | **49.9%** | **74.2%** | 8.5% | **58.8%** | 1.1% | 0.7% |
| | CW-$L_2$ ($\kappa = 0$) | 12.5% | 12.7% | 8.8% | **51.8%** | 1.1% | 8.8% |
| | DeepFool | 12.7% | 12.8% | 9.2% | **48.3%** | 1.2% | 1.3% |
| | PGD | 17.2% | 1.2% | 32.7% | **72.6%** | **42.5%** | 79.7% |

Table 7: Here, we show Target Training performance on original, non-adversarial samples. When Target Training is not using adversarial samples against attacks with no perturbation, Target Training even exceeds unsecured classifier accuracy on non-adversarial samples (Adversarial Training is not applicable here because it needs adversarial samples). When Target Training is using adversarial samples against attacks with perturbation, Target Training equals Adversarial Training performance.

| Attack used for adv. samples in training | CIFAR10 (84.3%) | | | MNIST (99.1%) | | |
|---|---|---|---|---|---|---|
| | Target Training | Adversarial Training | No Defense | Target Training | Adversarial Training | No Defense |
| none (against min. perturb. attacks ) | 86.7% | NA | 84.3% | 98.6% | NA | 99.1% |
| CW-$L_2$ ($\kappa = 40$) | 77.7% | 77.4% | 84.3% | 98.0% | 98.0% | 99.1% |
| PGD | 76.3% | 76.9% | 84.3% | 98.3% | 98.4% | 99.1% |
| FGSM($\epsilon = 0.3$) | 77.6% | 76.6% | 84.3% | 98.6% | 98.6% | 99.1% |

Table 8: Class output probabilities for Target Training on original, and adversarial samples from MNIST. Adversarial samples generated with CW-$L_2(\kappa = 0)$. Zero probability values and probability values rounded to zero have been omitted.

| | Original images | | | | | | | | | |
|---|---|---|---|---|---|---|---|---|---|---|
| Labels | 0 | 1 | 2 | 3 | 4 | 5 | 6 | 7 | 8 | 9 |
| 0 | 0.51 | | | | | | | | | |
| 1 | | 0.44 | | | | | | | | |
| 2 | | | 0.62 | | | | | | | |
| 3 | | | | 0.78 | | | | | | |
| 4 | | | | | 0.75 | | | | | |
| 5 | | | | | | 0.62 | | | | |
| 6 | | | | | | | 0.65 | | | |
| 7 | | | | | | | | 0.61 | | |
| 8 | | | | | | | | | 0.52 | |
| 9 | | | | | | | | | | 0.43 |
| 10 | 0.49 | | | | | | | | | |
| 11 | | 0.57 | | | | | | | | |
| 12 | | | 0.38 | | | | | | | |
| 13 | | | | 0.22 | | | | | | |
| 14 | | | | | 0.25 | | | | | |
| 15 | | | | | | 0.38 | | | | |
| 16 | | | | | | | 0.35 | | | |
| 17 | | | | | | | | 0.39 | | |
| 18 | | | | | | | | | 0.48 | |
| 19 | | | | | | | | | | 0.57 |

| | Adversarial images | | | | | | | | | |
|---|---|---|---|---|---|---|---|---|---|---|
| | 0 | 1 | 2 | 3 | 4 | 5 | 6 | 7 | 8 | 9 |
| 0 | 0.50 | | | | | | | | | |
| 1 | | 0.50 | | | | | | | | |
| 2 | | | 0.49 | | | | | | | |
| 3 | | | | 0.49 | | | | | | |
| 4 | | | | | 0.50 | | | | | |
| 5 | | | | | | 0.50 | | | | |
| 6 | | | | | | | 0.50 | | | |
| 7 | | | | | | | | 0.46 | | |
| 8 | | | | | | | | | 0.50 | |
| 9 | | | | | | | | | | 0.50 |
| 10 | 0.50 | | | | | | | | | |
| 11 | | 0.50 | | | | | | | | |
| 12 | | | 0.51 | | | | | | | |
| 13 | | | | 0.51 | | | | | | |
| 14 | | | | | 0.50 | | | | | |
| 15 | | | | | | 0.50 | | | | |
| 16 | | | | | | | 0.50 | | | |
| 17 | | | | | | | | 0.54 | | |
| 18 | | | | | | | | | 0.50 | |
| 19 | | | | | | | | | | 0.50 |

Table 9: Class output probabilities for Target Training on original, and adversarial samples from CIFAR10. Adversarial samples generated with CW-$L_2(\kappa = 0)$. Zero probability values and probability values rounded to zero have been omitted. The two highest class probabilities for each image are made bold. The deer (fifth image) appears to be misclassified as a horse.

| | Original images | | | | | | | | | |
|---|---|---|---|---|---|---|---|---|---|---|
| Labels | air-plane | auto-mobile | bird | cat | deer | dog | frog | horse | ship | truck |
| 0 | **0.41** | | | | | | | | | |
| 1 | | **0.46** | | | | | | | | |
| 2 | 0.01 | | **0.56** | | | | | | | |
| 3 | | | | **0.60** | | | | | | |
| 4 | | | | | 0.08 | | | | | |
| 5 | | | | | 0.01 | **0.48** | | | | |
| 6 | | | | | | | **0.53** | | | |
| 7 | | | | | **0.39** | | | **0.56** | | |
| 8 | | | | | | | | | **0.54** | |
| 9 | | | | | | | | | | **0.47** |
| 10 | **0.58** | | | | | | | | | |
| 11 | | **0.55** | | | | | | | | |
| 12 | 0.01 | | **0.43** | | | | | | | |
| 13 | | | | **0.40** | 0.01 | | | | | |
| 14 | | | | | 0.06 | | | | | |
| 15 | | | | | 0.01 | **0.52** | | | | |
| 16 | | | | | | | **0.47** | | | |
| 17 | | | | | **0.45** | | | **0.44** | | |
| 18 | | | | | | | | | **0.46** | |
| 19 | | | | | | | | | | **0.53** |

| | Adversarial images | | | | | | | | | |
|---|---|---|---|---|---|---|---|---|---|---|
| Labels | air-plane | auto-mobile | bird | cat | deer | dog | frog | horse | ship | truck |
| 0 | **0.49** | | | 0.01 | | | | | | |
| 1 | | **0.54** | | | | | | | | |
| 2 | 0.01 | | **0.49** | | | | | | | |
| 3 | | | | **0.47** | | | | | | |
| 4 | | | | | 0.08 | | | | | |
| 5 | | | | | 0.01 | **0.51** | | | | |
| 6 | | | | | | | **0.48** | | | |
| 7 | | | | | **0.44** | | | **0.49** | | |
| 8 | | | | | | | | | **0.46** | |
| 9 | | | | | | | | | | **0.50** |
| 10 | **0.49** | | | | | | | | | |
| 11 | | **0.46** | | | | | | | | |
| 12 | 0.01 | | **0.50** | | | | | | | |
| 13 | | | | **0.53** | 0.01 | | | | | |
| 14 | | | | | 0.05 | | | | | |
| 15 | | | | | 0.01 | **0.49** | | | | |
| 16 | | | | | | | **0.52** | | | |
| 17 | | | | | **0.41** | | | **0.51** | | |
| 18 | | | | | | | | | **0.54** | |
| 19 | | | | | | | | | | **0.50** |

