# OpenReview forum: "Target Training: Tricking Adversarial Attacks to Fail"
_ICLR.cc/2021/Conference — Reject_

### Official Review · AnonReviewer1 · 2020-10-27
**Not sure if this is an effective defense**

**Rating:** 5
**Confidence:** 5

**Review:**

This paper proposed an ad hoc defense mechanism against white-box attacks, by duplicating the training data with original samples or adversarial samples and the number of prediction classes. The authors claim that this method achieves better results compare to baseline methods.

I admit that this method could potentially defend against gradient-based attacks like CW and PGD if the attacker have no knowledge of the defense mechanism. However, since this method is defending the white-box  threat model, I believe a simple attack could break it:

To generate adversarial sample for test data $x$ whose correct label is $y$ or $y+k$, run PGD algorithm with the objective function $\hat{x}= argmax_{x} l(f(x),y)+l(f(x),y+k)$, where $\hat{x}$ is the adversarial example, $k$ is the number of classes (before duplication) and $f$ is the network trained by target training. Namely, this is just maximize the loss for both the real class $y$ and the duplicated class $y+k$.
I'm not sure why the authors did not include any adaptive attacks like this one in section 5.

Other flaws:

-The writing is confusing, a lot of details are omitted. For example, what is perturbation size for CIFAR10 under PGD attack?

-In section 4.1, the authors say "Thus, the undefeated Adversarial Training defense cannot be used as a baseline because
it uses adversarial samples during training for all types of attack." I don't buy it. The choice of baseline method should be based on the threat model, not the algorithm or training data used.

-The authors say "Adversarial training assumes that the attack is known..." I believe this is not true.

-Black-box attack ZOO shows only 81.5% accuracy on unsecured classifier, which basically means this is not an effective black-box attack. How could the authors use an ineffective attack to demonstrate the effectiveness of their defense method?

In summary, given the execution of the experiment, I'm not convinced this is an effective defense against white-box attacks.

---

> ### Public Comment · ~Abhay_Yadav1 · 2020-11-10
> **Not an effective defense for the an adaptive white-box attack**
>
> I agree with the reviewer that its not an effective defense against white box attack. And it turns out, it won't give you any significant edge over other white-box methods as well (verified empirically if that helps).

---

> ### Author Response · Authors · 2020-11-13
> **Target Training withstands the adaptive attack you suggest**
>
> Thank you for your valuable feedback! Please find below answers to your remarks.
>
> The attack you are suggesting was also suggested by AnonReviewer3. As I explained to them also, Target Training still works against this adaptive attack because what the attack achieves is to convert the Target-Trained classifier into an Adversarially-Trained classifier: the classifier that is changed becomes a classifier trained with nearby samples that are labeled correctly. Therefore, Target Training in this case would work for the same reason that Adversarial Training works.
>
> I have also implemented this adaptive attack and get:
> For Cifar10 - 86.56\% for DeepFool, 83.15\% for CarliniWagner($\kappa=0$), 75.63\% for CarliniWagner($\kappa=40$), 7.14\% for PGD.
> For Mnist - 94.88\% for DeepFool, 95.72\% for CarliniWagner($\kappa=0$), 93.92\% for CarliniWagner($\kappa=40$), 57.93\% for PGD.
> The results are similar to the results of transferability attacks, but with better accuracies except for the PGD attack.
>
> -We agree that more details of the attacks should have been included. CIFAR10 under PGD had: eps=8.0, eps_step=2.0, num_random_init=0, max_iter=7, same as the PGD paper by Madry et al for CIFAR10.
>
> -Adversarial Training is known to not approach original classifier, clean image accuracy, whereas Target Training exceeds it for adversarial attacks that minimize perturbation in CIFAR10 - for example 86.2\% for CarliniWagner($\kappa=0$).
>
> -In Adversarial Training, a classifier needs to be trained with adversarial samples of the attack that it is supposed to counter: at each iteration of the training. In order to generate the adversarial samples, the attack needs to be known during training. This is why we state that "Adversarial training assumes that the attack is known...". Occasionally, it can happen that when a classifier is trained with adversarial samples from one attack, it can provide protection against other, usually similar, attacks as well. In Table 6 in our paper's Appendix, we highlight such cases. However, there are no guarantees unless the correct attack is used. Even with the correct attack used in Adversarial Training, different parameter settings lead to worse results.
>
> -The ZOO attack is very slow to run, due to which even its authors report results for only 200 randomly sampled test images. We did the same, and reported the results thinking that the size of the sample might have skewed the result.

---

> ### Author Response · Authors · 2020-11-24
> **Changes in the rebuttal version of the paper**
>
> We have submitted a rebuttal version of our paper where we address the issues you raised:
>
> -We implement an adaptive attack where the attack adds an extra layer with 20 inputs and 10 outputs where output $i$ is the sum of inputs $i$ and $k$. We find that Target Training withstands this adaptive attack for non-$L_\infty$ attacks, and in MNIST even the CW-$L_\infty$ attack. We discuss that we think that the adaptive attack fails (for non-$L_\infty$ attacks) because the addition of the extra layer turns the classifier into an Adversarial-Trained classifier. We also discuss that we think Target Training performance against $L_\infty$ attacks is affected by the low capacity of the classifiers we use. The results are summarized in Table 4. The results in the abstract, introduction, and the conclusion have been updated with the adaptive attack results.
> This adaptive attack fails even when this extra-layer, what-has-become Adversarial-Trained classifier is trained with only original samples, and no adversarial samples, in attacks that minimize perturbation. Which means that it works even when the low-density areas have not been populated with adversarial samples. This hints that Adversarial Training might work for the same reason that Target Training works, and not vice-versa. This is discussed in the Conclusion.
>
> -We have now added PGD attack details for MNIST and CIFAR10 in the paper: $7$ steps of size $2$ with a total $\epsilon=8$ for CIFAR10, and $40$ steps of size $0.01$ with a total $\epsilon=0.3$ for MNIST. For all PGD attacks, we use $0$ random initialisations within the $\epsilon$ ball, effectively starting PGD attacks from the original images.
>
> -We have now added Adversarial Training as a baseline for attacks that minimize perturbation, and have updated Section 4.1 accordingly. Adversarial Training results are shown in Table 1. The blank ZOO results there are explained in the following point.
>
> -We have added more ZOO attack details in attack parameters in Section 4, as well as in Table 1. We included the ZOO attack in the results because the high 81.5% accuracy on unsecured classifier can be explained by the random selection of only 200 testing samples. This is based on the ZOO paper, where they do the same for untargeted attacks because the attack is very slow. Since Adversarial Training needs adversarial samples at each iteration and ZOO attack is very slow, we also do not conduct ZOO Adversarial Training experiments and leave the ZOO results blank in Table 1.
>
> -We still maintain that Adversarial Training assumes that the attack is known in advance, since it needs to generate adversarial samples during training.

---

### Official Review · AnonReviewer4 · 2020-10-27
**A simple and interesting trick to foul the attacker**

**Rating:** 7
**Confidence:** 2

**Review:**

Summary:
This paper proposed  a simple trick (doubling the number of output class) to foul the minimization, gradient-based adversarial attacks. The method is simple, powerful and requires nearly no changes to the current training infrastructure.  This paper provides a new direction how to design defense strategy.

Pros:
(1) This idea is interesting and even requires no changes during the training process. Also, the author provides enough intuitions why they design this based on the current model of adversarial attack.
(2) The experiment results are impressive and even exceeds unsecured classifier accuracy on non-adversarial samples.

Cons:
(1) The paper is not well written especially for the intro part (no more information is provided compared to their abstract).  Also, some formula need more explanations to help better understanding, e.g., loss_f in the minimization 1.
(2) The defense strategy is based on the minimization model of attacks which may restrict its generality.

Some typos:

(1) In the last paragraph of page 2, ...many gradient-based attacksSome.... This should be a typo.
(2) What's the difference between the two optimization problems in page 2 and 3 ?  They are the same. Also, please label them for
better reference.

---

> ### Author Response · Authors · 2020-11-14
> **Thank you**
>
> Thank you very much for your feedback and time!
>
> I'll make the suggested corrections.

---

> ### Author Response · Authors · 2020-11-24
> **Changes in the rebuttal version of the paper**
>
> We have submitted a rebuttal version of our paper where we address the issues you raised:
>
> - We have now added an explanation for loss_f, and removed $l$ from the Minimization formula to avoid confusion in untargeted attacks.
>
> - We think that the fact that many attacks are based on this minimization makes our attack widely applicable.
>
> - We have corrected the typo, thanks for pointing it out.
>
> - We have removed the second minimization formula as we thought it might add to confusion without much added benefit.
>
> In addition:
>
> We implement an adaptive attack suggested by another reviewer where the attack adds an extra layer with 20 inputs and 10 outputs where output $i$ is the sum of inputs $i$ and $k$. We find that Target Training withstands this adaptive attack for non-$L_\infty$ attacks, and in MNIST even the CW-$L_\infty$ attack. We discuss that we think that the adaptive attack fails (for non-$L_\infty$ attacks) because the addition of the extra layer turns the classifier into an Adversarial-Trained classifier. We also discuss that we think Target Training performance against $L_\infty$ attacks is affected by the low capacity of the classifiers we use. The results are summarized in Table 4. The results in the abstract, introduction, and the conclusion have been updated with the adaptive attack results.
> This adaptive attack fails even when this extra-layer, what-has-become Adversarial-Trained classifier is trained with only original samples, and no adversarial samples, in attacks that minimize perturbation. Which means that it works even when the low-density areas have not been populated with adversarial samples. This hints that Adversarial Training might work for the same reason that Target Training works, and not vice-versa. This is discussed in the Conclusion.

---

### Official Review · AnonReviewer2 · 2020-10-27
**The idea of target training is novel, but several key issues on potential limitations are not evaluated**

**Rating:** 5
**Confidence:** 3

**Review:**

Summary:
This paper proposes target training to defend against adversarial attacks on machine learning models. Target training doubles the number of output classes, and aims to trick untargeted attacks into attacks that target at designated classes. Experimental results show that targeting training can achieve slightly better performance than adversarial training.

Pros:
1. Target training applies simple changes to the structure of machine learning models, to defend against adversarial attacks.
2. The idea of tricking adversarial attacks is novel.
3. Target training can partially break transferability of adversarial samples.

Cons:
1. What is the overhead of target training?
2. Would target training degrade the performance on clean data?
3. More discussion should be included on whether target training is effective against adaptive attacks.


Detailed comments:
While the idea of applying target training to defend against adversarial attacks is interesting, I have the following questions regarding the proposed method (performance, limitation, etc.).

1. Target Training aims to convert “untargeted attacks to attacks targeted at designated classes”, but doesn’t Minimization 1 in Section 2.1 correspond to targeted attacks rather than untargeted attacks (l is the target label)?

2. What is the overhead of target training, especially Algorithm 3? How does that compare with normal training and adversarial training?

3. Adversarial training degrades the model’s performance on clean data. Does target training have the same limitation?

4. Would Algorithm 1 (Algorithm 3) also be working against attacks that do not (do) minimize perturbations? How to choose between these two algorithms?

5. Not effective against adaptive attacks is one of the main limitations of many existing defence method. It is unclear whether target training has the same limitation. Not using techniques that have been broken does not mean that target training is robust.

---

> ### Author Response · Authors · 2020-11-13
> **Answers to your questions**
>
> Thank you for your valuable feedback! Please, find below the answers to your questions.
> 1. The Minimization 1 is based on the first formulation of this Minimization by Szegedy et al. (cited in the paper). The $l$ in this minimization is a variable, not a constant. $l$ can be any of the adversarial labels. On the other hand, the $c$ in the minimization is introduced as a constant, right after $l$.
> 2. In Algorithm 3, Target Training (TT) and Adversarial Training (AT) have the same overhead over normal training. Both TT and AT need to generate adversarial samples in each iteration of the training. This slows down the training considerably because the classifier in its current state has to be used to generate adversarial samples in each batch.
> In Algorithm 1 against attacks that minimize perturbation, TT does not use adversarial samples, so this overhead is no longer there for TT.
> 3. For attacks the minimize perturbation in CIFAR10, TT improves accuracy on clean data to 86.7\% from 84.3\%. For other attacks, it has the same limitation as AT. These results are shown in Table 5 in the Appendix.
> 4. Table 6 highlights cases where training with samples from attacks that do not minimize perturbation can sometimes offer defense against attacks that minimize perturbation (CarliniWagner($\kappa=0$), DeepFool). This happens more in MNIST than in CIFAR10, and more in TT than in AT, as shown in Table 6. For example, when trained with CarliniWagner($\kappa=40$) adversarial samples, TT achieves 71.3\% on CarliniWagner($\kappa=0$), whereas AT achieves 13.2\%.
> Algorithm 1 and Algorithm 3 need not be exclusive of each-other (Algorithm 3 + Algorithm 3 can also be combined to protect against two types of attacks). For Algorithm 1 + Algorithm 3 combination, we would concatenate the batches of Alg. 1 and Alg. 3, and the top probabilities would be about $0.25$. This could even be extended further for more attacks. I have not implemented any of this. Another approach, used in AT, would be randomly selecting in each batch the attack(s) used for generation of adversarial samples.
> 5. AnonReviewer3 and AnonReviewer1 have suggested an adaptive attack that we think TT is able to withstand. The adaptive attacker, they suggest, could add an extra layer on top of the classifier with 20 inputs and 10 outputs, where every output $i$ is the sum of inputs $i$ and $i+k$. We argue that this would effectively turn the Target-Trained classifier into an Adversarially-Trained classifier because the classifier has been trained with additional samples that are close to the original samples and which are labeled correctly (due to the relabeling from the extra layer). Against this adaptive attack, TT would work for the same reason that AT works.
> I have implemented this adaptive attack and get:
> For Cifar10 - 86.56\% for DeepFool, 83.15\% for CarliniWagner($\kappa=0$), 75.63\% for CarliniWagner($\kappa=40$), 7.14\% for PGD.
> For Mnist - 94.88\% for DeepFool, 95.72\% for CarliniWagner($\kappa=0$), 93.92\% for CarliniWagner($\kappa=40$), 57.93\% for PGD.
> The results are similar to the results of transferability attacks, but with better accuracies except for the PGD attack.

---

> ### Author Response · Authors · 2020-11-24
> **Changes in the rebuttal version of the paper**
>
> We have submitted a rebuttal version of our paper where we address the issues you raised:
>
> 1. We have now removed $l$ from the Minimization formula to avoid confusion in untargeted attacks.
>
> 2. As stated in the previous reply, in Algorithm 3, Target Training (TT) and Adversarial Training (AT) have the same overhead over normal training. Both TT and AT need to generate adversarial samples in each iteration of the training. This slows down the training considerably because the classifier in its current state has to be used to generate adversarial samples in each batch.
>
> In Algorithm 1 against attacks that minimize perturbation, TT does not use adversarial samples, so this overhead is no longer there for TT. The overhead stays for AT.
>
> 3. For attacks the minimize perturbation in CIFAR10, TT improves accuracy on clean data to 86.7\% from 84.3\%. For other attacks, it has the same limitation as AT. These results are now shown in Table 7 in the Appendix.
>
>
> 4. As stated in the previous reply, Table 6 highlights cases where training with samples from attacks that do not minimize perturbation can sometimes offer defense against attacks that minimize perturbation (CarliniWagner($\kappa=0$), DeepFool). This happens more in MNIST than in CIFAR10, and more in TT than in AT, as shown in Table 6. For example, when trained with CarliniWagner($\kappa=40$) adversarial samples, TT achieves 71.3\% on CarliniWagner($\kappa=0$), whereas AT achieves 12.3\% (correction from previous reply).
>
> We have now added in the paper that Target Training could be extended to defend simultaneously against many attacks by having a designated class for each attack, and training with adversarial samples labeled with the designated class. However, we also state in the paper that we do not conduct experiments for such multi-attack defense in this paper.
>
> 5. We implement an adaptive attack where the attack adds an extra layer with 20 inputs and 10 outputs where output $i$ is the sum of inputs $i$ and $k$. We find that Target Training withstands this adaptive attack for non-$L_\infty$ attacks, and in MNIST even the CW-$L_\infty$ attack. We discuss that we think that the adaptive attack fails (for non-$L_\infty$ attacks) because the addition of the extra layer turns the classifier into an Adversarial-Trained classifier. We also discuss that we think Target Training performance against $L_\infty$ attacks is affected by the low capacity of the classifiers we use. The results are summarized in Table 4. The results in the abstract, introduction, and the conclusion have been updated with the adaptive attack results.
> This adaptive attack fails even when this extra-layer, what-has-become Adversarial-Trained classifier is trained with only original samples, and no adversarial samples, in attacks that minimize perturbation. Which means that it works even when the low-density areas have not been populated with adversarial samples. This hints that Adversarial Training might work for the same reason that Target Training works, and not vice-versa. This is discussed in the Conclusion.

---

### Official Review · AnonReviewer3 · 2020-10-28
**Not sure if the evaluation is valid**

**Rating:** 5
**Confidence:** 3

**Review:**

This paper addresses the task of adversarial defense, particularly against untargeted attack. It starts from the observation that these attacks mostly minimize the perturbation and the classification loss, and proposes a new training strategy named Target Training. The method duplicate training examples with a special ground-truth label, to fool the adversarial attackers. Experiments are conducted on MNIST and CIFAR10 under several attacks.

\+ The proposed method is simple, and involves a little modification to the classifiers and small computation overheads.

\- In the bottom of page 5, authors claimed that "adversary has complete knowledge of how the defense works". Could the authors please elaborate that, and how are the attacks modified to handle the proposed defense? If the untargeted attack is conducted on a 20-way classifier (for MNIST) without modification, I am not sure this evaluation is valid. One can add an extra layer after the last layer of the classifier, mapping $(y_0,\cdots, y_{2k-2})$ to $(y_0+y_k,\cdots,y_{k-1}+y_{2k-1})$, and then perform attacks -- since this is how the classifier actually works.

\- In Equation Minimization 1, there is a tern $l$ which is the adversarial label. I am not sure it is appropriate here since we are doing untargeted attacks.

\- From Table 2, the proposed method seems to achieve almost the same performance compared with vanilla adversarial training.

\- In Section 4.3, authors say Target Training maintains performance on clean images, when trained with adversarial samples. This might be inaccurate -- we can say Target Training achieves same performance on clean images compared with vanilla adversarial training.

---- Post rebuttal ---
I appreciate the responses from the authors, which partially address the concerns I was having. However, I am still not fully convinced that the proposed method is significant enough. Thus I am increasing my rating from 4 to 5.

---

> ### Author Response · Authors · 2020-11-10
> **Suggested adaptive attack already evaluated in transferability analysis**
>
> Thank you for your valuable feedback! Please, find below answers to your minus (-) remarks.
>
> -The kind of adaptive attack that the reviewer suggests has already been evaluated in the transferability evaluation, Table 3. There, the adversarial samples are generated with a 10-way unsecured classifier. That 10-way classifier does not have the 20-way extra layer of the reviewer-suggested classifier, but the 10-way classifier should be harder to defeat because it is closer to the original architecture of the classifier (it has no extra layer). We agree that we should have mentioned this in the adaptive evaluation.
>
> -Equation Minimization 1 is based on the formulation by Szegedy et al. in the referenced paper.
>
> -We agree, Table 2 also states the same. The point of Table 2 was to show that Target Training maintained performance similarly to Adversarial Training on attacks that do not minimize perturbation.
>
> -You are right, except for one case: when Target-Trained for attacks that minimize perturbation in CIFAR10, the classifier accuracy on clean images exceeds default classifier performance: 86.7% (up from 84.3%).

---

> > ### Comment · AnonReviewer3 · 2020-11-10
> > **White-box adaptive attack**
> >
> > If I understand correctly, Table 3 is for transferability analysis, not the white-box adaptive attack. What I suggested is when attacking a target training model $F$ with 20-way outputs, we extend the model $F$ by an extra layer $m$ whose inputs are $(y_0,y_1,\cdots,y_{2k-1})$ and outputs are $(y_0+y_k,\cdots,y_{k-1}+y_{2k-1})$. Then the attack is applied on the extended model $G=m\odot F$. In Table 3, the 10-way classifier is not an extended version of $F$, but something else. Please correct me if I was wrong. Thanks!

---

> > > ### Author Response · Authors · 2020-11-11
> > > **Target Training withstands the adaptive attack you suggest**
> > >
> > > Thank you again for your feedback and time!
> > >
> > > First, to clarify, the 10-way classifier in Table 3 is not an extended version of $F$.
> > >
> > > Target Training works against the adaptive attack you suggest because what the attack achieves is to convert the Target-Trained $F$ classifier into the Adversarially-Trained $G$ classifier. $G$ is now a classifier trained with nearby (or exactly the same) samples that are labeled correctly. Therefore, Target Training in this case works for the same reasons that Adversarial Training works.
> > >
> > > I have also implemented the attack you suggest and get:
> > > In Cifar10 - 86.56\% for DeepFool, 83.15\% for CarliniWagner($\kappa=0$), 75.63\% for CarliniWagner($\kappa=40$), 7.14\% for PGD.
> > > In Mnist - 94.88\% for DeepFool, 95.72\% for CarliniWagner($\kappa=0$), 93.92\% for CarliniWagner($\kappa=40$), 57.93\% for PGD.
> > >
> > > The results are similar to the results of transferability attacks, but with better accuracies except for the PGD attack.

---

> ### Author Response · Authors · 2020-11-24
> **Changes in the rebuttal version of the paper**
>
> We have submitted a rebuttal version of our paper where we address the issues you raised:
>
> - We have implemented the adaptive attack you suggest: where the attack adds an extra layer with 20 inputs and 10 outputs where output $i$ is the sum of inputs $i$ and $k$. We find that Target Training withstands this adaptive attack for non-$L_\infty$ attacks, and in MNIST even the CW-$L_\infty$ attack. We discuss that we think that the adaptive attack fails (for non-$L_\infty$ attacks) because the addition of the extra layer turns the classifier into an Adversarial-Trained classifier. We also discuss that we think Target Training performance against $L_\infty$ attacks is affected by the low capacity of the classifiers we use. The results are summarized in Table 4. The results in the abstract, introduction, and the conclusion have been updated with the adaptive attack results.
> This adaptive attack fails even when this extra-layer, what-has-become Adversarial-Trained classifier is trained with only original samples, and no adversarial samples, in attacks that minimize perturbation. Which means that it works even when the low-density areas have not been populated with adversarial samples. This hints that Adversarial Training might work for the same reason that Target Training works, and not vice-versa. This is discussed in the Conclusion.
>
> - We have now removed $l$ from the Minimization formula to avoid confusion in untargeted attacks.
>
> - We have stated in the paper, including in Table 2, that Target Training maintained performance similarly to Adversarial Training on attacks that do not minimize perturbation.
>
> - We have now clarified in Section 4.3 that Target Training performance on original samples is maintained and exceeded when trained without adversarial samples against atacks that minimize perturbation; whereas Target Training original sample performance when trained with adversarial samples against attacks that do not minimize perturbation is comparable to Adversarial Training.

---

### Decision · Program_Chairs · 2021-01-07
**Final Decision**

**Decision:**

Reject

**Comment:**

I thank the authors and reviewers for the lively discussions. Reviewers found the work to be interesting but some concerns were raised regarding the significance of the results. In particular, two reviewers mentioned that authors did not fully address their concerns in the rebuttal period. Given all, I think the paper still needs a bit of work before being accepted. I recommend authors to address comments raised by the reviewers to improve their work.

-AC